

# Reconstructed SqueezeNext with C-CBAM for offline handwritten Chinese character recognition

Ruiqi Wu, Feng Zhou, Nan Li, Xian Liu and Rugang Wang

School of Information Technology, Yancheng Institute of Technology, Yancheng, Jiangsu, China

## ABSTRACT

**Background**. Handwritten Chinese character recognition (HCCR) is a difficult problem in character recognition. Chinese characters are diverse and many of them are very similar. The HCCR model consumes a large number of computational resources during runtime, making it difficult to deploy to resource-limited development platforms.
**Methods**. In order to reduce the computational consumption and improve the operational efficiency of such models, an improved lightweight HCCR model is proposed in this article. We reconstructed the basic modules of the SqueezeNext network so that the model would be compatible with the introduced attention module and model compression techniques. The proposed Cross-stage Convolutional Block Attention Module (C-CBAM) redeploys the Spatial Attention Module (SAM) and the Channel Attention Module (CAM) according to the feature map characteristics of the deep and shallow layers of the model, targeting enhanced information interaction between the deep and shallow layers. The reformulated intra-stage convolutional kernel importance assessment criterion integrates the normalization nature of the weights and allows for structured pruning in equal proportions for each stage of the model. The quantization aware training is able to map the 32-bit floating-point weights in the pruned model to 8-bit fixed-point weights with minor loss.
**Results**. Pruning with the new convolutional kernel importance evaluation criterion proposed in this article can achieve a pruning rate of 50.79% with little impact on the accuracy rate. The various optimization methods can compress the model to 1.06 MB and achieve an accuracy of 97.36% on the CASIA-HWDB dataset. Compared with the initial model, the volume is reduced by 87.15%, and the accuracy is improved by 1.71%. The model proposed in this article greatly reduces the running time and storage requirements of the model while maintaining accuracy.

# INTRODUCTION

The written word is a symbolic system used by humans to communicate, a form of writing to record ideas and time (*Wang et al., 2018a*; *Wang et al., 2018b*). As a unique tool for recording, preserving, and transmitting knowledge, Chinese characters have played a great role in the advancement of human civilization in China and the world, and in promoting the progress and development of Chinese society.

Corresponding author
Feng Zhou, zfycit@ycit.edu.cn

HCCR is the recognition of already written Chinese characters as characters in the standard character set (*Liu et al., 2022*). HCCR tasks can be divided into two types: online and offline. Online recognition tasks are mainly used in scenarios such as terminal handwriting input. Online recognition models can record strokes and trajectories, and the presence of noise is less influential. Compared to online tasks, offline recognition tasks are a difficult area in character recognition (*Yan & Wang, 2017*; *Jin et al., 2016*). At present, offline handwritten Chinese character recognition technology is mainly used in areas such as bank cheque recognition, electronic paper archives, handwritten manuscript recognition, and marking homework in primary and secondary schools (*Shen & Messina, 2016*). Equipment such as copiers and scanners with text recognition functions can save a lot of time and greatly improve work efficiency. Online recognition models and recognition models for printed fonts are numerous. These two types of models are easier to implement and have higher accuracy rates. Offline recognition models and handwriting recognition models have to cope with more complex noise and complicated input images and are therefore relatively slow to develop. The scenarios where offline handwritten character recognition tasks need to be performed are mostly devices with limited computing resources such as scanners and sorters.

The model needs to be compressed for better-embedded platform development. By compressing convolutional neural networks through techniques such as reconstruction (*Howard et al., 2017*; *Zhang et al., 2018*), quantization (*Chen et al., 2015*), pruning (*Han, Mao & Dally, 2015*; *Luo et al., 2019*; *Li et al., 2016*), knowledge distillation (*Hinton, Vinyals & Dean, 2017*), and low-rank decomposition (*Denton et al., 2014*). These techniques not only speed up training, convergence, and forward propagation, but also reduce the computational resources consumed at runtime.

After depth-wise separable convolution was proposed, most lightweight models adopt this type of convolution, leading to a slowdown in the development of importance assessment algorithms for ordinary convolution and lightweight architectures such as SqueezeNet (*Iandola et al., 2016*) and SqueezeNext (*Gholami et al., 2018*). The implementation of depth-wise separable convolution on hardware platforms does not perform as well as it does on GPUs. Traditional field-programmable gate array (FPGA) accelerators are designed with different IP cores for computation acceleration of point-wise convolution and channel-wise convolution separately. (*Wu et al., 2019*; *Ding et al., 2019*). However, embedded FPGA resources in edge devices are often limited, making it impractical to support separate hardware designs for channel-wise convolution and element-wise convolution (*Xie et al., 2023*).

As research in convolutional neural network technology progresses, the network models applied to HCCR are also evolving. The number of parameters and the size of the model are constantly being increased to improve the accuracy of the model's recognition. The increase in model performance also brings with it some drawbacks. Models take up a lot of computing resources and consume a lot of power when they are running. The increase in the number of parameters also makes training difficult, makes optimization difficult, and takes up a lot of storage space. This can cause many inconveniences for deployment on resource-constrained platforms such as mobile and hardware (*Hua, 2020*).

The compression efficiency for offline handwritten Chinese character recognition models is insufficient. Fewer developers have adopted lightweight model architectures, and the compression measures applied to the models, while effective, are not very helpful for deployment to hardware platforms.

In order to effectively address the problems with the HCCR model, the following targeted improvements have been made in this article.

- Reconstruction of the structure of the SqueezeNext (hereafter referred to as SqNxt) network basic block. Compared to the initial block, the reconstructed block has a slightly higher volume. Correspondingly, the recognition accuracy of the model has been improved. The adapted feature fusion approach also gives the conditions for model compression such as pruning. A high-accuracy lightweight network architecture containing only ordinary convolution or spatial separable convolution for the task of offline handwritten Chinese character recognition is constructed.
- Introduction of a cross-stage attention module. This attention module enhances both shallow and deep semantic information interaction. The internal structure of the attention module is adapted to the different characteristics of the deep and shallow feature maps, preserving the spatial information of the shallow tensor and the channel information of the deep tensor.
- A new criterion for assessing the importance of convolutional kernels is proposed. L1 and L2 paradigms are introduced and configured with different weights. Pruning with this criterion can compress the model volume without compromising accuracy. Combined with quantization, a good overall compression effect can be achieved.

## RELATED WORKS

Most of the current research in Chinese character recognition is aimed at improving the accuracy. *Qin, Zheng & Zhang (2020)* proposed a model which first performs classification recognition employing a quadratic discriminant function and determines whether a deep Boltzmann machine is required for secondary recognition based on the generalized confidence level. *Li et al. (2020)* designed a method to enrich feature information through feature mixing and washing, feature grouping extraction, and re-fusion, which can achieve a high recognition accuracy. However, the network structure is complex and resource consumption is high. *Melnyk, You & Li (2020)* used a ''bottleneck layer'' and global weighted output averaging pooling and reduced the parameters by 49% while maintaining a high recognition accuracy at the same computational cost. *Xu (2022)* introduced an attention model to the GoogLeNet and improved classification accuracy to 98.1%.

In order to obtain higher accuracy rates, models are becoming larger and more redundant. This makes model compression a necessity. Model compression of deep neural networks can be beneficial for various application scenarios. Pruning and quantization is one of the more widely used model compression techniques as it is not constrained by factors such as model structure. After evaluating the importance of neurons, *Liu et al. (2023)* pruned unimportant neurons through Huffman coding compression iteration, and then quantized weights using K-means++ clustering. This approach reduced the memory

footprint and FLOPs while maintaining accuracy. *Zhuo et al. (2023)* designed a knowledge transfer-based data-free post-training quantization method, which achieved high accuracy even when the model weights and activation values were compressed to six bits. It efficiently implemented detection tasks. To accelerate the federated learning process on edge devices, *Ren et al. (2023)* performed channel pruning and weight quantization on the model to reduce communication and storage costs.

Parameters can be compressed, both in terms of the number (pruning) and the space occupied by one parameter (quantization). Current pruning techniques consist mainly of structured pruning and unstructured pruning. Structured pruning targets channels or convolutional filters and removes a channel or a convolutional filter directly. Unstructured pruning (*Han et al., 2015*) is relatively more fine-grained, targeting connections. While this approach compresses the model while maintaining accuracy, the resulting sparse matrix requires the support of special algorithm libraries for deployment to hardware platforms. The irregular sparsity is difficult to accommodate in general-purpose hardware architectures and typically requires customized hardware design (*Ma et al., 2021*). For example, *Song et al. (2022)* designed a dedicated accelerator for sparse matrix–vector multiplication based on high-bandwidth memory, which achieved a memory-centric processing engine capable of supporting sparse matrix–vector multiplication. However, it heavily relies on high-bandwidth and large-capacity on-chip memories, making it challenging to apply to edge computing platforms.

The compression effect of pruning is heavily influenced by the importance of the evaluation criteria used. *Li et al. (2017)* proposed measuring importance by the absolute magnitude of convolutional kernel weights and determining pruning ratios for each layer in the neural network through sensitivity analysis. *He et al. (2019)* proposed using the geometric median of the convolutional kernels at each layer for pruning. The geometric median is defined as the value that minimizes the distance from the norm of each convolutional kernel's weight. The article suggests that the closer the norm value is to the geometric median, the less important the corresponding convolutional kernel is, and thus should be pruned, rather than following the principle of smaller norm values implying lesser importance. *Hu et al. (2016)* proposed evaluating neurons based on activation layers and introduced the concept of average percentage of zeros (APoZ). A higher APoZ indicates that the corresponding neuron is less important. Based on the same idea, *Liu et al. (2017)* proposed an importance assessment method based on the output of the batch normalization (BN) layer. They introduced scaling factors in the channels and obtained sparsity factors through penalty terms. After determining the pruning ratio, the channels with smaller scaling factor values will be pruned. *Sakai et al. (2022)* proposed an automatic pruning rate derivation method to reduce the inefficient workload of manually assigning pruning rates. For various types of ResNet models on CIFAR-10 and ImageNet, this method achieved the highest compression ratios in terms of parameters and FLOPS. The pruning ratio optimizer (PRO) proposed by *Kamma, Inoue & Wada (2022)*, combined with reconstruction error aware pruning (REAP), achieves layer-wise pruning effectiveness.

The quantization replaces the high-precision 32-bit weights with low-precision 16-bit or 8-bit ones. This approach not only reduces the storage space required for the model but also

reduces the computing time required to run the model. The quantization can quantize only the weights (*Zhou et al., 2017*), or the weights and activation values (*Courbariaux et al., 2016*; *Wang et al., 2018a*; *Wang et al., 2018b*; *Krishnamoorthi, 2018*; *Cai et al., 2018*; *Jacob et al., 2017*). Three quantization schemes (uniform quantizer, uniform symmetric quantizer, and stochastic quantizer) and two quantization algorithms (post training quantization and quantization aware training) were proposed in the literature (*Krishnamoorthi, 2018*).

Currently, there are few studies related to lightweight HCCR. The model proposed by *Zho, Tan & Xi (2021)* achieved an accuracy of 96.32% on the ICDAR-2013 dataset and compressed the model from 4.7MB to 3.2MB by using a dynamic network pruning algorithm to prune the parameters of SqueezeNet (hereafter referred to as SqNt) and re-patch important connections. Using MobileNetV3 as the basis, *Cheng et al. (2022)* used multi-scale convolutional kernels to improve the accuracy to 96.68% while keeping the number of parameters low.

## Basic module redefinition

The SqueezeNext network is made up of multiple iterations of the basic module. In this article, multiple basic modules are treated as a stage. Convolutional layers for down-sampling are interspersed between stages. Therefore, the performance of the basic module can directly affect the performance of the overall model. In the original model, the Add function was used for feature fusion. However, the Add function requires not only the same feature resolution but also the same number of channels of the two sets of feature tensors to be fused, *i.e.,* the same number of convolution kernels. This limits the number of model compression methods that can be used to adjust the number of convolutional kernels, such as pruning. The reconstructed basic module proposed in this article is shown in Fig. 1.

The proposed basic module in this article uses two bottleneck layers, BottleNeck1 and BottleNeck2, for channel count compression. The two convolutional layers originally connected in series using K*1 and 1*K convolutional kernels are replaced with parallel ones. The output of Bottleneck 2 will be passed into these layers at the same time. The output of these two convolutional layers will be fused with features by concatenating. The concatenated feature maps will then be concatenated again with the input features. The number of channels of the tensor obtained by concatenation is large, so it needs to go through another bottleneck to adjust the number of channels. The feature concatenating approach increases the number of parameters but retains all the output of the previous convolutional layers and there is no limit to the number of channels. Conditions are created for model compression methods such as pruning.

To adapt the SqNxt structure to the handwritten Chinese character dataset, the filter size of the first convolutional layer in the original network is changed from 7*7 to 3*3 and the stride is reduced to 1. The backbone undergoes three times resolution compression to obtain four sets of feature maps with different resolutions. The number of repetitions for basic modules is two, four, 14, and one. There are 3,755 Chinese characters in the dataset, so the number of channels needs to be adjusted to avoid a sudden expansion in the number of channels leading to a large amount of redundant information.

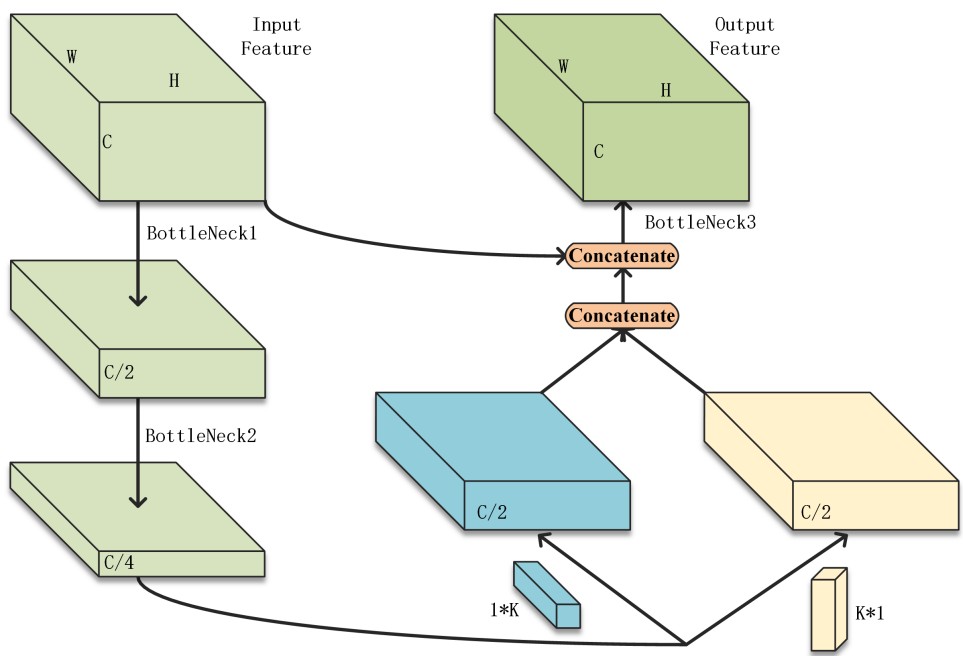

**Figure 1** The restructured basic module.

## Enhanced attention module: C-CBAM

Chinese characters are pictographs. Many Chinese characters are so similar that an additional stroke is all that is needed to form another character. In this article, we introduce an enhanced attention module C-CBAM in addition to the optimization basic module. C-CBAM, which is an improved version of CBAM (*Woo et al., 2018*), consists of two sub-modules, CAM and SAM.

First, referring to the idea of ECA (*Wang et al., 2020*), the fully connected layer in CAM is replaced by Conv1D and the layer used for channel scaling is removed, which reduces the number of parameters in the CBAM module and improves operational efficiency while maintaining accuracy. The filter size of the Conv1D layer is calculated in the same way as ECA, as shown in Eq. (1), where C is the number of input channels.

$$filter\_size = (\log_2 C + 1)/2 \tag{1}$$

With a traditional CBAM attention module, CAM is first applied to the input features to obtain the channel modulation features and then SAM is applied to obtain the final output features. However, as SAM is applied to the channel modulation features, the effect of the SAM is influenced by the CAM. At the shallow end of the model, the number of feature maps is small and the spatial information of the features is not compressed. The spatial information at this point is of greater importance to the overall model. Therefore, the order of the two sub-modules will be the same as the original CBAM, ensuring that the feature map is spatial modulated before output. In the middle layer of the model, the two modules are deployed in parallel. The two sets of modulation features are fused and output. In the deeper layers of the model, the number of channels expands several times,

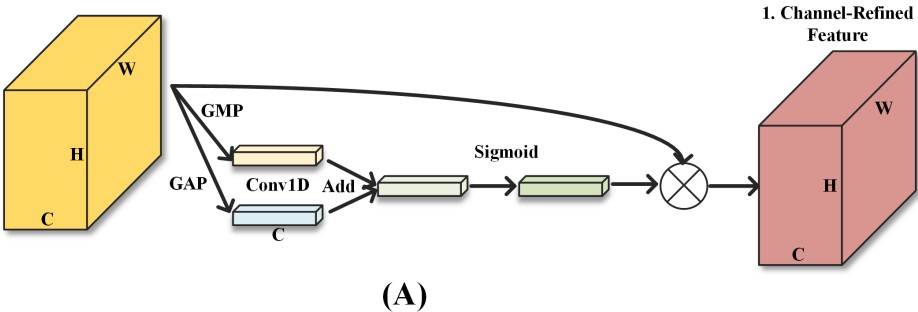

**(A)**

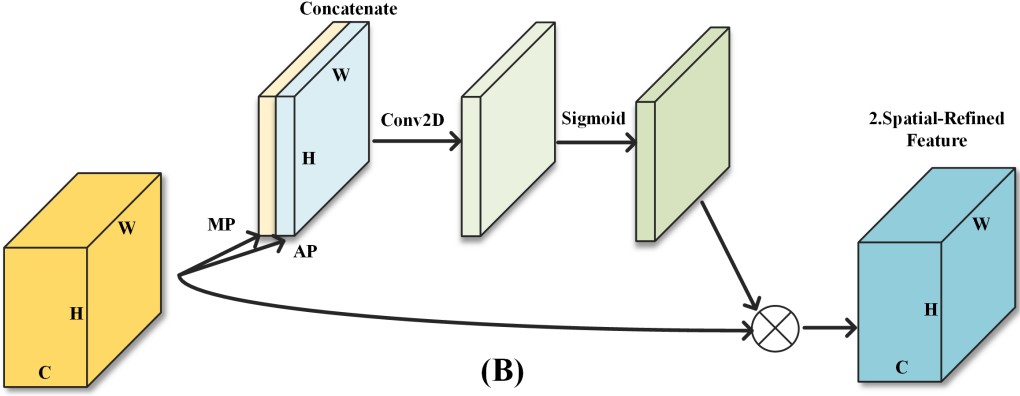

**(B)**

**Figure 2** **The two sub-modules of the enhanced attention module.** (A) Sub-module 1: Channel attention module; (B) Sub-module 2: Spatial attention module.

while the spatial information is highly compressed. The channel information at this point then has a greater impact on the classification accuracy of the Softmax function. Moving the SAM module in CBAM before the CAM module ensures that the output features are finally modulated by channel attention. The two sub-modules are shown in Fig. 2.

The attention module is deployed in the method shown in Fig. 3. Where K3 indicates a convolutional filter size of 3, S1 indicates a stride of 1 and C64 indicates a number of output channels of 64. Modules 1, 2, and 3 in the dashed box indicate three different structures of CBAM with CAM in front, CAM and SAM in parallel, and SAM in front respectively. The output of ConvFirst is used as input to C-CBAM1. The output of the MaxPooling layer before Block3 is used as input to C-CBAM2. The output of the MaxPooling layer before Block7 is used as input to C-CBAM3. This residual connectivity across multiple basic modules can improve the interaction between shallow and deep information and services to enhance deep semantic information.

The convolutional layer is initialized with 'he_normal'. To maintain good classification accuracy at low precision (16 or 8 bits) and to reduce the impact of subsequent quantization operations on the accuracy, the activation function used in this article is ReLU. The h-swish activation function proposed in MobileNetV3 is only advantageous in deep networks so that ReLU will be able to satisfy the requirements of the network we proposed.

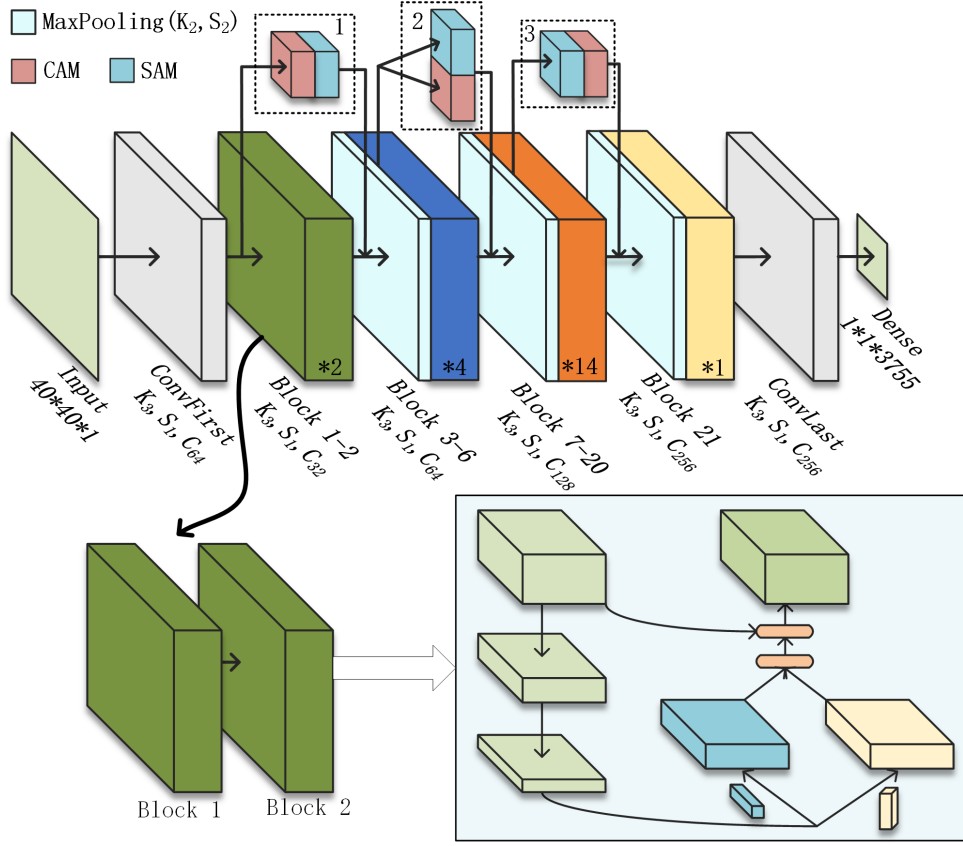

**Figure 3  Employing the C-CBAM.**

The proposed C-CBAM can enhance detection accuracy, but it does not assist with model compression. The module has a small number of parameters, which slightly increases the model's size and computational requirements.

## Pruning

The most straightforward method of model compression is to reduce the number of parameters, that is the number of convolutional filters. In addition to adjusting the number of convolutional filters per layer directly, network pruning can also be done to reduce the number of convolutional filters. Network pruning can be divided into structured pruning and unstructured pruning. Structured pruning refers to the removal of the complete convolutional filter. The structured pruning process is shown in Fig. 4. Unstructured pruning refers to fine-grained pruning, where some weights are removed from the convolutional filter, but not the entire filter. Unstructured pruning is highly flexible and can form sparse networks without sacrificing accuracy. However, pruning the weights can make the matrix too sparse, leading to redundant memory allocations that affect the operational efficiency of the network, and when deployed with the hardware side, it relies on algorithm libraries such as cuSPARSE, making it difficult to accelerate parallel
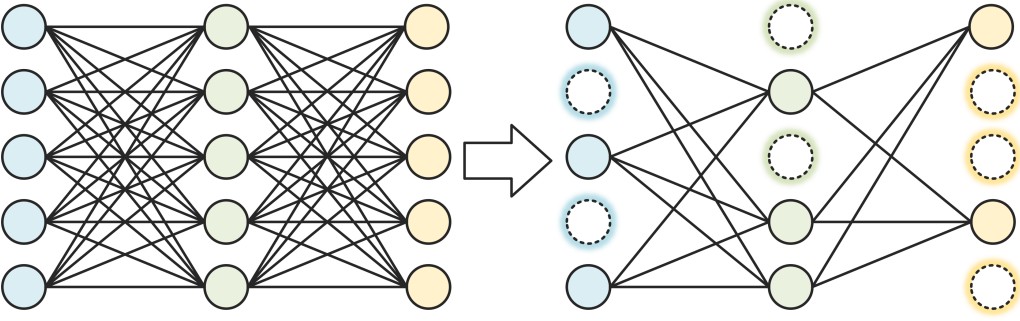

**Figure 4  Structured pruning process.**

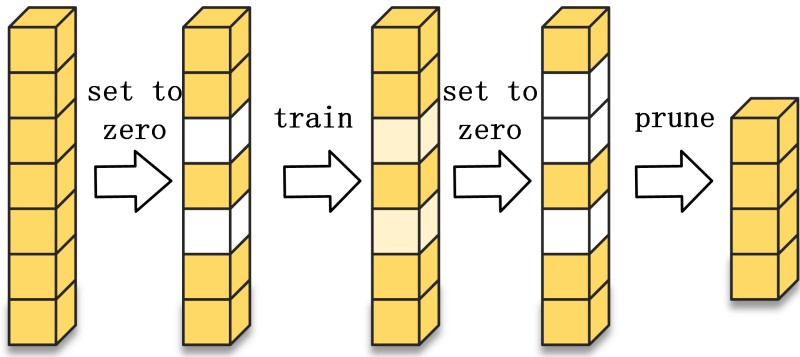

**Figure 5  Asymptotic soft filter pruning process.**

operations through the circuit. Due to the limitations of unstructured pruning, structured pruning is used in this article.

The pruning process uses asymptotic soft filter pruning (ASPF). The technique is characterized by resetting to zero, rather than removing, the filters that satisfy the cropping criteria during training. These filters will continue to participate in subsequent training. The number of filters that are set to zero is then gradually expanded. At the end of the last training epoch, this part of the convolution filter is pruned. Compared to pruning the convolutional filters directly during training, ASPF gradually focuses the information on the more important convolutional filters. The ASFP process is illustrated in Fig. 5.

The standard of pruning is an important factor in the effectiveness of pruning. Pruning operations that are based directly on the magnitude of the sum of the absolute values of the convolutional kernel weights can cause large fluctuations in accuracy. In this article, we introduce the L1 and L2 norms in the proposed importance evaluation formula, as shown in Eq. (2). where $\alpha$ and $\beta$ are scaling adjustment factors and $\theta$ is the weight.

$$Ipt = \alpha \sum_{i=1}^{n} \sum_{j=1}^{n} |\theta_{ij}| + \beta \sqrt{\sum_{i=1}^{n} \sum_{j=1}^{n} \theta_{ij}^2} \tag{2}$$

When the values of the elements in the matrix change significantly, the fluctuations in the L1 norm of the matrix are usually larger and the fluctuations in the L2 norm are smaller. So the L1 and L2 norm of each matrix will be weighted and then summed. When the L1 norm is too large, the matrix is too sparse or the element values are too small, the model may be under-fitted and the sparse weights are not conducive to subsequent model compression means; when the L1 norm is too small, the feature extraction performance of the weights is reduced and useful information cannot be filtered out. When the L2 norm is too large, it pulls the weights to a small value, resulting in an underfitting of the model; when it is too small, it loses the effect of smoothing the curve. Therefore, $\alpha$ and $\beta$ need to be set within a reasonable range to ensure the validity of the evaluation formula.

As the convolutional kernel weights may vary significantly between the shallow and deep layers of the model, in order to obtain a standardised wide range of thresholds that are not affected by outliers, the evaluation formula requires the values of all convolutional kernels in the full model by counting them. Compared to global thresholding, a multiple-stage thresholding approach allows a more appropriate threshold to be established based on the specific distribution of weights in each stage between each downsampled layer, ensuring that there will not be a case of cropping only one part of the model. In this article, the $Z$-score normalisation method is used to indicate the importance of a value in that stage, as shown in Eq. (3),

$$Ipt_{stage}^{i} = \frac{Ipt^{i} - \overline{Ipt}}{Ipt_{\sigma}} \tag{3}$$

where $\overline{Ipt}$ and $Ipt_{\sigma}$ represent the mean and standard deviation of all values in a single stage, respectively. The importance of the convolution kernel is determined by ranking the $Ipt_{stage}^{i}$. The i-th convolutional kernel corresponds to importance $Ipt^{i}$ and is ordered within the stage $Ipt_{stage}^{i}$. This normalisation method is designed to limit the threshold to a single stage and to facilitate pruning without affecting the original sorting order of $Ipt_{stage}^{i}$. Half of the values after normalisation are less than zero. During the experiment, not all convolution kernels corresponding to a less than zero are pruned.

Pruning is carried out according to the following steps and the flow chart is shown in Fig. 6.

Repeat:

1. The model before this round of pruning is noted as $M_0$.
2. Zero the weights in X% of the convolution kernels in the model that meet the clipping criteria to obtain the $M_1$.
3. Compare the accuracy of $M_0$ and $M_1$.

   - If $\Delta acc(M_0 - M_1) \geq 1.0\%$, it means that the accuracy fluctuates too much after the greedy algorithm has set the weight of that part of the convolution kernel to zero, and this round of pruning needs to be abandoned.

4. $M_1$ is fine-tuned to recover accuracy after several training sessions to obtain $M_2$
5. Compare the accuracy of $M_0$ and $M_2$.

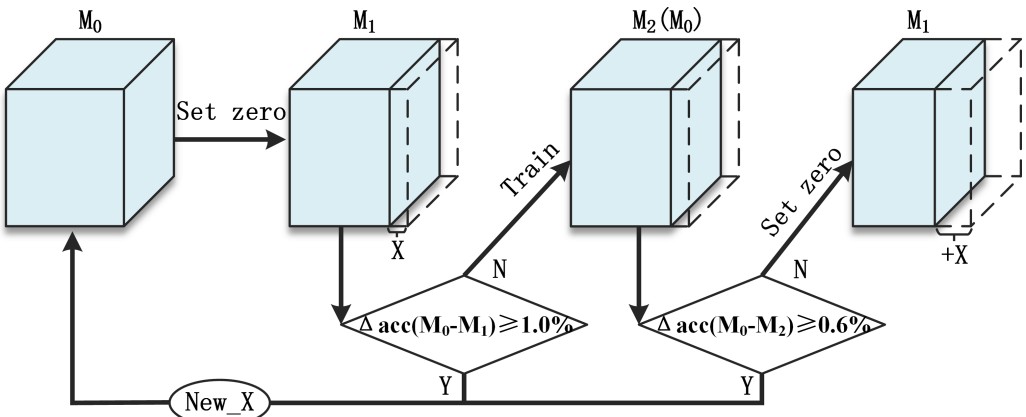

**Figure 6** Overall pruning process.

- If $\Delta acc(M_0 - M_2) \geq 0.6\%$, it means that the pruning has cut the more important convolutional kernels and the accuracy fluctuates a lot, and the accuracy cannot be recovered by training fine-tuning, so this round of pruning needs to be abandoned

6. If $\Delta acc(M_0 - M_1) \leq 1.0\%$ and $\Delta acc(M_0 - M_2) \leq 0.6\%$, then this pruning has minimal impact on the accuracy of the model and can be trained to improve accuracy and is in a position to continue with the next round of pruning. At this point $M_2$ is used as $M_0$ for the next round of pruning. Due to the soft pruning, the number of parameters and the structure of the model do not change at this point. At the start of a new pruning round, an additional X% of the convolution kernels are set to zero on top of step 2.

7. If this pruning round is abandoned at step 3 or 5, the $M_0$ model from step 1 needs to be read and the size of X and the $Ipt^i_{stage}$ criteria adjusted downwards to start the next pruning round with a smaller growth ratio.

Following the above steps for structured pruning allows redundant convolutional kernels to be quickly screened out and zeroed at the beginning of the pruning. The increment $X$ is decreasing and the number of convolutional kernels to be zeroed at a time is increasing. The number of convolutional kernels for cropping is shown in Eq. (4), where $k_i$ denotes the number of pruning rounds at increments of $X_i$.

$$\delta K = \sum_{i=1}^{N} k_i X_i \tag{4}$$

## Quantization

The main quantization methods for TensorFlow models are post-training quantization and quantization-aware training. Post-training quantization is used for floating-point models that have already been trained. Quantization-aware training is the calculation of simulated low-precision numerical types during the training of the model, which can compensate for the decrease of accuracy caused by quantization during the training process and speed up

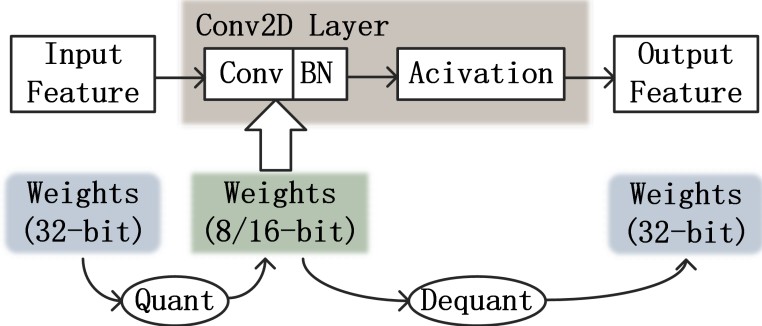

**Figure 7** Simulating low precision processes in CNN.

forward propagation. It can approximate the accuracy achieved by the float32 model as closely as possible, enabling deployment on hardware platforms.

To simulate low-precision calculations, a fake-quantization operation needs to be introduced into the training. The simulated low-precision computational flow is shown in Fig. 7. After obtaining the feature map of the upper layer input, the weights of the 32-bit are quantized to low-precision values. The low precision weights will be convolved with the feature map and then the output of this convolution layer will be obtained by ReLU activation. At this time, the weights will be inversely quantized to a 32-bit floating point type.

Fake-quantization nodes are added to the computation process of each layer to count the minimum and maximum values of float32 data for each weight at training time. The fake-quantization nodes are not involved in backpropagation, as gradient updates need to be calculated under the floating-point type. Accurate gradient updates require exact weights. Between the two convolution layers, there will be an inverse quantization and a quantization operation adjacent to each other. After the weights are back-quantized back to the floating point at the output of the upper layer, they are quantized again to low precision values without any operation. In this case, a lot of computational resources are wasted. Therefore, when inverse quantization and quantization are adjacent, they can cancel each other out and no operation is performed on the weights.

As each value in the floating-point type tensor maps one-to-one to a low-precision value, any further computation using the tensor does not introduce additional losses and can accurately simulate low-precision computations. The quantized simulation operations need to be integrated into the training process to be consistent with the quantized computation.

As symmetric quantization is not effective in representing unevenly distributed fractions, this article uses asymmetric quantization, where the mapping of zeros to floating point numbers requires an offset. The mapped scale factor $P$ is shown in Eq. (5), where $F_{\max}$ and $F_{\min}$ denote the maximum value of the 32-bit floating point number counted, respectively, and $I_{\max}$ and $I_{\min}$ denote the maximum value that can be represented by int8, respectively.

$$P = \frac{F_{\max} - F_{\min}}{I_{\max} - I_{\min}} = \frac{F_{\max} - F_{\min}}{255} \tag{5}$$

Once the proportional relationship is obtained, it can be mapped to the corresponding int8 integer value based on the 32-bit floating point value, as shown in Eq. (6), where $I$ denotes an int value, $F$ denotes a float value, and the $round()$ function denotes rounding. $Z$ represents the integer value to which the floating-point zero value is mapped, as shown in Eq. (7). When symmetric quantization is used, $P * I_{max} = F_{max}$, the floating-point number $Z$ corresponds to the integer type zero.

$$I = round(\frac{F}{P} + Z) \tag{6}$$

$$Z = round(I_{max} - \frac{F_{max}}{P}) \tag{7}$$

Based on the quantization process of individual values, the quantization needs to be extended to the multiplication and addition operations of the convolution kernel. Unlike the determinant calculation, the convolution operation involves multiplying the corresponding positions of the matrix and finally adding all the products to obtain the final result. The quantization process for multiplying two floating point values is shown in Eq. (8):

$$P_3(I_3^{i,j} - Z_3) = P_1(I_1^{i,j} - Z_1)P_2(I_2^{i,j} - Z_2) \tag{8}$$

The expression for $I_3^{i,j}$ is obtained by collation, as shown in Eq. (9):

$$I_3^{i,j} = \frac{P_1 P_2}{P_3}(I_1^{i,j} - Z_1)(I_2^{i,j} - Z_2) + Z_3 \tag{9}$$

In Eq. (9), all values are integers, except for the scale factor $(P_1 P_2)/P_3$.

Assuming that the size of the convolution kernel is L*L, the single convolution operation is shown in Eq. (10).

$$M_F = \sum_{i=0,j=0}^{L} m_3^{i,j} = \sum_{i=0,j=0}^{L} m_1^{i,j} m_2^{i,j} \tag{10}$$

where $M_F$ denotes the value obtained from a single convolution operation, $m_1$, $m_2$ and $m_3$ denote the matrix to be convolved and the resultant matrix respectively, all with 32-bit floating-point values in the matrix. Bringing Eq. (8) into the matrix yields a fixed-point to floating-point formula for each value, as shown in Eq. (11).

$$M_I = \sum_{i=0,j=0}^{L} P_3(I_3^{i,j} - Z_3) = \sum_{i=0,j=0}^{L} P_1(I_1^{i,j} - Z_1)P_2(I_2^{i,j} - Z_2) \tag{11}$$

## Dataset and experimental settings

The dataset was selected from the offline handwritten Chinese character dataset of the Institute of Automation, Chinese Academy of Sciences (CASIA-HWDB), with 3,755 commonly used characters at the first level of the GB2312 standard. As shown in Fig. 8, some samples from the dataset are depicted, with the text in the image representing "Offline Handwritten Chinese Character Recognition."

There are miswriting in the dataset as well as scribbles after the miswriting. In addition, some of the writing fonts are too shallow and the resolution of the parsed individual

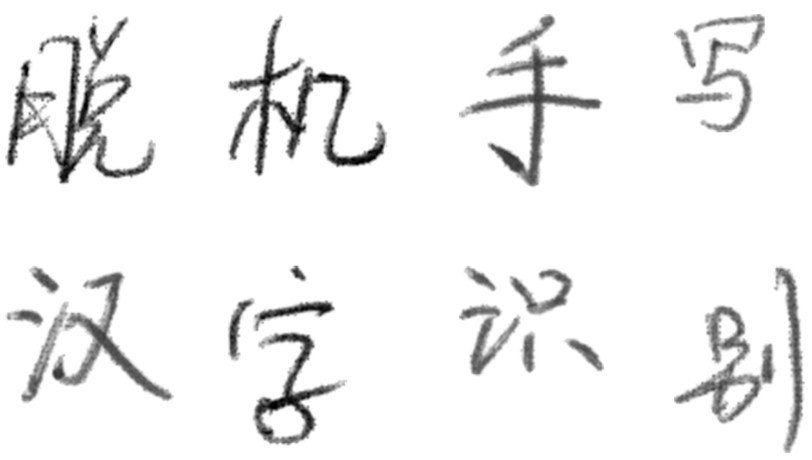

**Figure 8** Examples of the dataset.

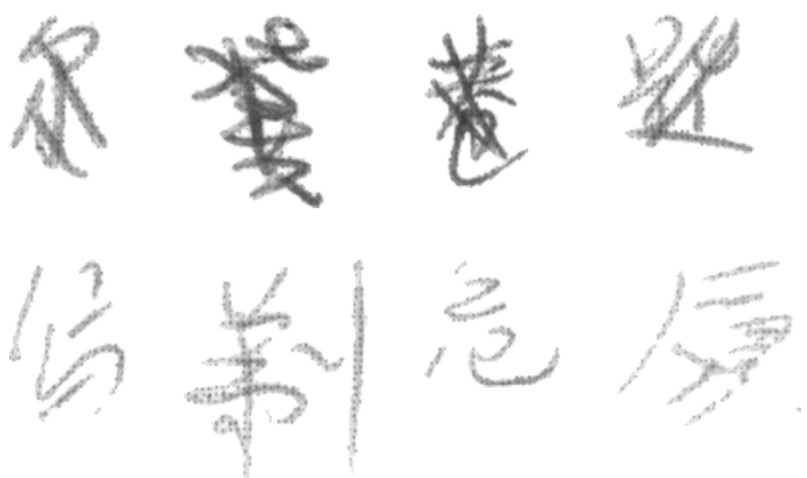

**Figure 9** The samples need to be processed.

characters varies, as shown in Fig. 9, so the dataset needs to be pre-processed for normalization. All samples were binarised and stored using the average grey level of the full image as the threshold. Samples after preprocessing and normalization are shown in Fig. 10.

The data from HWDB 1.0 and HWDB 1.1 were mixed and washed to exclude as many useless samples as possible. Finally, 400 samples of each character were selected for training and 100 samples for testing. The details are shown in Table 1.

This experiment is based on Tensorflow-GPU version 2.3.0, using a GPU with NVIDIA RTX2060 6G memory, Intel Core i5-9400@2.9 GHz CPU, and DDR4 2667MHz 16G+16G memory.

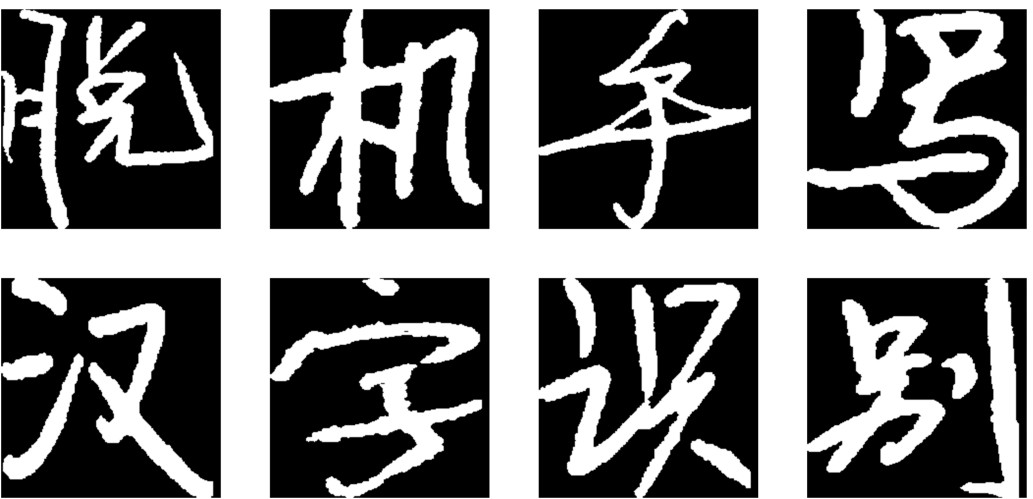

**Figure 10** Samples after preprocessing and normalization.

**Table 1** Dataset details.

| Dataset | Classification | Writers | | Total samples | | Selected samples |
|---------|----------------|---------|------|---------------|------|------------------|
| | | Train | Test | Train | Test | |
| HWDB 1.0 | 3740 | 336 | 40 | 1,246,991 | 309,684 | 1,502,000 |
| HWDB 1.1 | 3755 | 240 | 60 | 897,758 | 223,991 | 375,500 |

## RESULTS AND DISCUSSION

To verify the effectiveness of the various improvement strategies proposed, a series of ablation experiments were carried out. The network models were trained using the same settings in the framework of experiments on the same platform. The impact of the proposed improvement modules on the performance of the network model, as well as the improvement effect of multiple modules acting together, is verified separately in the test set. The results are shown in Table 2. The Roman numerals I - IV in the table indicate each of the four proposed improvements: I. basic module redefinition; II. deployment of an enhanced attention module; III. pruning based on module redefinition; and IV. quantification.

In the table, Scenario 0 represents the baseline model.

Scenario 1: In addition to the changes to the basic module structure, the number of convolutional kernels in the structure has also been adjusted. Block 7–20, which has the highest number of basic modules, has received more parameters. As can be seen from the data in the table above, the re-architecture of the basic block achieved a very good optimization. The new module will increase the size of the model by 3.88% and, accordingly, the accuracy of the recognition has been improved by 0.83%. The specific changes in the number of parameters between Scenario 0 and Scenario 1 are shown in Table 3.

**Table 2  Ablation study results for proposed strategies.**

| No. | I | II | III | IV | Acc (%) | Model size (/MB) | GFLOPs |
|---|---|---|---|---|---|---|---|
| 0 | | | | | 95.65 | 8.25 | 0.033 |
| 1 | ✓ | | | | 96.48 (+0.83) | 8.57 | 0.035 |
| 2 | | ✓ | | | 96.69 (+1.04) | 8.35 | 0.034 |
| 3 | ✓ | | ✓ | | 96.40 (+0.75) | 4.06 | 0.017 |
| 4 | | | | ✓ | 95.59 (−0.06) | 2.14 | / |
| 5 | ✓ | ✓ | ✓ | ✓ | 97.36 (+1.71) | 1.06 | 0.017 |

**Table 3  Specific changes in parameters difference between Scenario 0 and Scenario 1.**

| Layers | Scenario 0 parameters | Scenario 1 parameters | Parameters difference | Reduction rate (%) |
|---|---|---|---|---|
| Conv_First | 832 | 832 | 0 | 0 |
| Block 1-2 | 14,752 | 16,032 | 1,280 | +8.68 |
| DownSample_1 | 4,224 | 4,224 | 0 | 0 |
| Block 3-6 | 40,960 | 80,672 | 39,712 | +96.95 |
| DownSample_2 | 8,320 | 8,320 | 0 | 0 |
| Block 7-20 | 534,272 | 1,168,960 | 634,688 | +118.79 |
| DownSample_3 | 16,512 | 16,512 | 0 | 0 |
| Block 21 | 491,008 | 195,968 | −295,040 | −60.09 |
| Conv_Last | 590,336 | 295,424 | −294,912 | −49.96 |
| FC | 484,395 | 484,395 | 0 | 0 |
| Total | 2,185,611 | 2,271,339 | 85,728 | +3.92 |

Scenario 2: C-CBAM improved accuracy by 1.04%, reaching 96.69%. The effectiveness of this attention module is remarkable. As far as the number of statistical parameters is concerned, the sum of the number of parameters of the three modules is very small. The rest of the increase in the number and volume is brought about by the residual connections. To validate the effectiveness of the proposed attention deployment order, further ablation experiments were conducted based on the three deployment positions and three different order arrangements depicted in Fig. 3. The results are shown in Table 4, where I, II, and III represent the three different deployment positions, and ①, ②, and ③ represent CAM in the front, CAM and SAM in parallel, and SAM in the front, respectively. From the data in the table, it can be observed that incorporating attention modules can improve the detection accuracy of the model to varying degrees. The approach adopted in this article achieves the optimal results.

Scenario 3: The concatenated feature fusion used in the new basic module ensures that pruning is feasible. This scenario is therefore based on top of Scenario 1. The specific changes in the number of parameters between Scenario 0 and Scenario 3 are shown in Table 5. Again, the fully connected layer still retains a large number of parameters. Block 21

**Table 4  Ablation study results on different orderings of attention modules.**

| No. | Position | | | Acc (%) |
|---|---|---|---|---|
| | I | II | III | |
| 0 | | | | 95.65 |
| 1 | ① | ① | ① | 96.31 (+0.66) |
| 2 | ② | ② | ② | 96.22 (+0.57) |
| 3 | ③ | ③ | ③ | 96.38 (+0.73) |
| 4 | ③ | ② | ① | 96.29 (+0.64) |
| 5 | ① | ② | ③ | 96.69 (+1.04) |

**Table 5  Specific changes in parameters difference between Scenario 0 and Scenario 3.**

| Layers | Scenario 0 parameters | Scenario 3 parameters | Parameters difference | Reduction rate (%) |
|---|---|---|---|---|
| Conv_First | 832 | 832 | 0 | 0 |
| Block 1–2 | 14,752 | 4,751 | −10,001 | −67.79 |
| DownSample_1 | 4,224 | 2,176 | −2,048 | −48.49 |
| Block 3–6 | 40,960 | 18,737 | −22,223 | −54.26 |
| DownSample_2 | 8,320 | 4,096 | −4,224 | −50.77 |
| Block 7–20 | 534,272 | 262,925 | −271,347 | −50.79 |
| DownSample_3 | 16,512 | 8,064 | −8,448 | −51.16 |
| Block 21 | 491,008 | 171,471 | −319,537 | −65.08 |
| Conv_Last | 590,336 | 276,992 | −313,344 | −53.08 |
| FC | 484,395 | 484,395 | 0 | 0 |
| Total | 2,185,611 | 1,234,439 | −951,172 | −43.52 |

and the last layer of convolution have a high reduction rate, also thanks to the optimization adjustments in Scenario 1.

Scenario 4: The quantization techniques have a much better compression ratio for model size, reaching 74.01%. Quantification does not produce changes to the number of parameters. The detection speed is also substantially improved.

Scenario 5: This scenario combines all of the above optimization scenarios. The size of the model was reduced by 87.15% to a final size of 1.06 MB, and the accuracy was also greatly improved, reaching 97.74% accuracy on the publicly available dataset CASIA-HWDB. Both high classification accuracy and high running speed are achieved.

A comparison of the model size and the time required to traverse the test set is shown in Fig. 11. The speed increase in forward propagation from pruning and quantization is significant. However, the reduction in traversal time is not as high as the proportional reduction in the volume of the model. The change in the bit-width of the parameters reduces the size of the model by nearly three-quarters, but the runtime reduction does not reach the same proportion. This is because, for high-performance GPUs, even 32 bits can

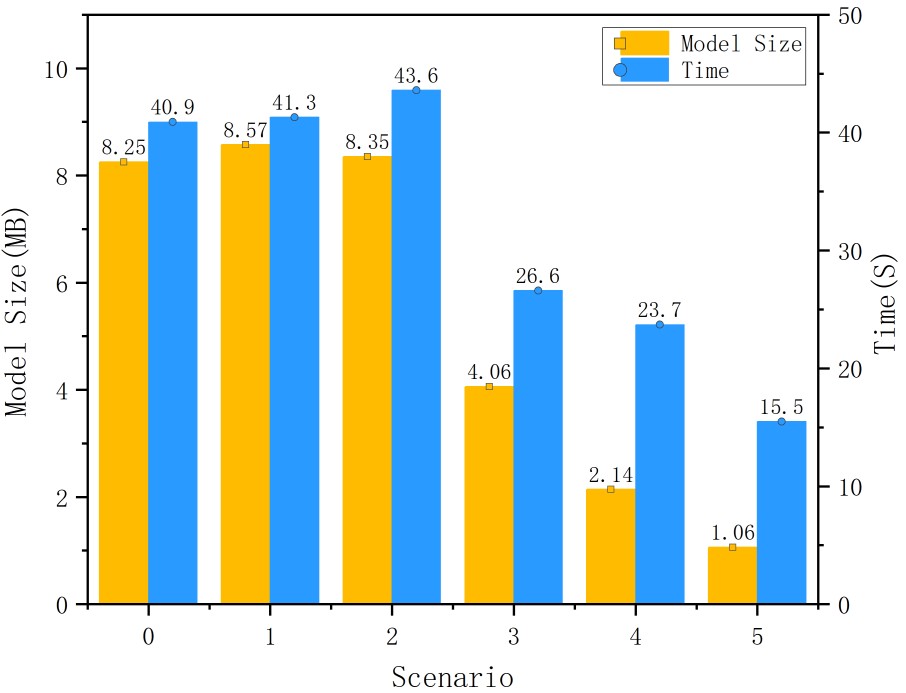

**Figure 11** Comparison of model size and running times of each scenario.

**Table 6** Comparative analysis of different activation functions' performance.

| Activation | Acc (%) | Time (s) |
|---|---|---|
| Sigmoid | 95.33 | 39.8 |
| Swish | 95.58 | 45.3 |
| H-Swish | 95.66 | 48.2 |
| Leaky-ReLU | 95.61 | 40.1 |
| ReLU | 95.65 | 40.9 |

achieve high computational speeds relative to 8 and 16-bit operations. The advantages of quantization can be seen on the GPU, but are not very obvious.

To verify the applicability of activation functions, comparative experiments were conducted on various activation functions. The results are shown in Table 6. From the results in the table, it can be observed that when using H-Swish, the model achieves the highest accuracy, but it also requires the longest time to traverse the dataset. On the other hand, when using the ReLU function, although the accuracy is reduced by 0.01%, the required time is significantly reduced. Therefore, considering the practical performance of each activation function, adopting ReLU as the activation function for the model in this article is the optimal choice.

Table 7 presents a comparison of results from state-of-the-art offline handwritten Chinese character recognition models in recent years. In the table, the asterisk (*) denotes models that have undergone lightweight processing or adopt lightweight architectures.

**Table 7  Comparison of different methods.**

| Method | Year | Acc (%) | Model size (MB) | GFLOPs | Parameters (Million) |
|---|---|---|---|---|---|
| HCCR-GoogLeNet | 2019 | 96.3 | N/A | N/A | N/A |
| Improved GoogLeNet | 2020 | 97.48 | N/A | N/A | N/A |
| 2DPCA+ACNN | 2020 | 93.63 | N/A | N/A | N/A |
| *SqueezeNet+DNS | 2021 | 96.32 | 3.2 | 1.58 | N/A |
| MSCS+ASA+SCL | 2022 | 97.63 | 22.9 | 0.18 | 6.01 |
| *HCCR-MobileNetV3 | 2022 | 96.68 | 3.87 | 0.06 | 5.86 |
| *LW-ViT | 2023 | 95.8 | 1.95 | 0.22 | 0.48 |
| *This study | 2023 | 97.36 | 1.06 | 0.017 | 1.23 |

**Notes.**

*Models that have undergone lightweight processing or adopt lightweight architectures.

From Table 7, it can be observed that the model designed in this article achieves the highest accuracy among the lightweight models. Additionally, compared to other models, it has a smaller size and requires less computational resources.

# CONCLUSIONS

In this article, we proposed a lightweight offline handwritten Chinese character recognition model. The model is based on the SqueezeNext network, with the basic modules reconstructed. The reconstructed base module increases the volume of the model slightly by 3.88% and improves the accuracy of recognition by 0.83%, while creating the conditions for structured pruning and C-CBAM. On the baseline model, the C-CBAM deployed on the residual side was able to increase the accuracy of the model to 96.69%. C-CBAM improves recognition accuracy by 1.04% with a volume increment of 1.21%. Compared with the network model reconstructed by the base module, structured pruning with the proposed convolutional kernel importance assessment algorithm in this article would reduce the recognition accuracy by 0.08%. At the same time, the volume of the model is reduced by 50.79%. Quantizing the weights from 32 bits to eight bits will reduce the model by about 3/4 of the volume as predicted. By combining the above optimizations, the optimal model was trained to achieve a classification accuracy of 97.36% on CASIA-HWDB with a model of only 1.06MB. Compared to the initial model, the accuracy has improved by 1.71%, the model size has been reduced by 87.15%. The overall performance has been greatly improved.

However, there are areas for improvement in the method proposed in this article. (1) Due to a large number of samples in the dataset, there are still problematic samples in the dataset, despite the pre-processing. Such wrong samples do not help to improve the performance of the model and can even lead to misclassification of the model. This also leads to difficulties in improving the accuracy of the model.

(2) Platforms such as ZYNQ are fast in parallel computing, but the process of scheduling and interacting with information on the AXI is time-consuming. While approaches such as low-rank decomposition can reduce the amount of computation and the number of

parameters, whether they can compensate for the delay caused by multiple information interactions requires further experimental verification.

### Funding
This research was funded by the Graduate Research and Innovation Projects of Jiangsu Province under grant number SJCX21 1517 and SJCX22 1685, the Major Basic Research Project of the Natural Science Foundation of the Jiangsu Higher Education Institutions under grant number 19KJA110002, the Natural Science Foundation of China under grant number No. 61673108 and the Yancheng Institute of Technology High level Talent Research Initiation Project under grant number XJR2022001. The funders had no role in study design, data collection and analysis, decision to publish, or preparation of the manuscript.

### Grant Disclosures
The following grant information was disclosed by the authors:
Graduate Research and Innovation Projects of Jiangsu Province: SJCX21 1517, SJCX22 1685.
Major Basic Research Project of the Natural Science Foundation of the Jiangsu Higher Education Institutions: 19KJA110002.
Natural Science Foundation of China: 61673108.
Yancheng Institute of Technology High level Talent Research Initiation Project: XJR2022001.

### Competing Interests
The authors declare there are no competing interests.

### Author Contributions
- Ruiqi Wu conceived and designed the experiments, performed the experiments, analyzed the data, performed the computation work, prepared figures and/or tables, authored or reviewed drafts of the article, and approved the final draft.
- Feng Zhou analyzed the data, authored or reviewed drafts of the article, and approved the final draft.
- Nan Li performed the experiments, analyzed the data, authored or reviewed drafts of the article, and approved the final draft.
- Xian Liu conceived and designed the experiments, performed the experiments, performed the computation work, authored or reviewed drafts of the article, and approved the final draft.
- Rugang Wang analyzed the data, prepared figures and/or tables, authored or reviewed drafts of the article, and approved the final draft.

### Data Availability
Raw data are available at the CASIA Online and Offline Chinese Handwriting Database: http://www.nlpr.ia.ac.cn/databases/handwriting/download.html.

Tag files generated for this article are available at Zenodo:

Ruiqi Wu. (2022). HCCR. https://doi.org/10.5281/zenodo.7445439.

## Supplemental Information

Supplemental information for this article can be found online at http://dx.doi.org/10.7717/peerj-cs.1529#supplemental-information.

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
