# Peer review of "Reconstructed SqueezeNext with C-CBAM for offline handwritten Chinese character recognition"

_PeerJ Computer Science, doi:10.7717/peerj-cs.1529_

## Round 0.1 · original submission · Major Revisions

Please consider reviewer comments to revise and resubmit the paper.

Reviewer 1 ·

Basic reporting

The authors present an enhanced lightweight HCCR model to improve computational efficiency. This is achieved through reconstruction of SqueezeNext's basic modules for pruning compatibility and the introduction of the Cross-stage Convolutional Block Attention Module (C-CBAM) to improve model performance.

1. The article is written in professional-level English that is easy to understand. However, there are some minor grammatical errors and typos that need to be corrected.

2. My main concern with the paper is the lack of proper citations. The author mentions various works, such as SqueezeNext and CAM, SAM, and CBAM, without properly citing them, some of these are only cited in the related works. Additionally, some abbreviations, such as ECA, are not explained, which makes it challenging to follow the text. The author should consider adding proper citations and explanations to make the text more accessible.

3. While the background, related work, and context to the problem are well-written and provide a good understanding of the problem statement, the sections on pruning and quantization lack citations for general things. It would be helpful to add some references to support the arguments presented in these sections.

4. One major issue with the paper is that the figures and tables are placed at the end, and the captions are not correctly placed. This makes it challenging to read and understand the paper. The author should consider placing the figures and tables near the relevant text and adding proper captions to improve the readability of the paper.

Experimental design

1. The paper proposes to redesign the Basic Module of the SqueezeNext Model to make it compatible with model compression techniques. The experiment is well-explained, and the results demonstrate an improvement in accuracy in this setting.

2. a) The paper proposes C-CBAM, which incorporates the CAM (channel attention module) and SAM (spatial attention module) in the modified SqueezeNext architecture. The motivation for using different orders of SAM and CAM is clearly explained in the paper. However, it is not clear if this block was specifically included for the purpose of compression. The authors should explain if this modified architecture also helps in compression.

b) Additionally, there appears to be a possible typo at line 202, where the authors write that the SAM module is placed after the CAM module so that output features are modulated by channel attention. This should be the other way around.

c) Validation of the proposed method: The results demonstrate increased accuracy compared to basic models. However, to validate the effectiveness of the suggested orders of CAM and SAM, authors should perform an ablation study while keeping the order the same in the middle layer as well as the end layers, and then comparing the accuracy with the proposed architecture.

3. The paragraph (lines 215-219) is not clear about how using the ReLU activation function would be advantageous over the h-swish activation function. The authors should perform an experiment or cite an article that supports this claim.

3. It would be beneficial if the authors included an end-to-end architecture figure that shows both modified blocks and C-CBAM together.

4. The authors have mentioned a few prior works related to model compression in HCCR. However, there is no table that performs a comparison with existing methods, which shows obtained accuracy versus the number of parameters. The authors should include such a table to better contextualize their results.

5. The authors have proposed a new criterion for evaluating the significance of convolutional kernels, which involves the introduction of L1 and L2 paradigms with varying weights. While the contributions are commendable, it is necessary for the authors to provide further comments and insights regarding the existing alternatives to their proposed method. Additionally, it would be beneficial to readers if the authors include a table showing the results obtained using their proposed criterion, to provide a comparative analysis of the effectiveness of the new method.

Validity of the findings

1. The study appears to be replicable and presents potential benefits for Chinese character recognition in low-resource settings. However, to enhance its utility, the authors could provide comparisons with existing methods.

2. The dataset used in the research is publicly accessible.

3. The study's conclusions are clearly articulated, and the authors have appropriately acknowledged current limitations.

4. The authors have excluded the proposed C-CBAM module from the code, presumably because of the ablation experiments being conducted. It would be helpful if the authors could comment on this decision.

Reviewer 2 ·

Basic reporting

The paper should improve English writing.

Experimental design

• Line no 152: incomplete sentences to be checked
• Justification for the selection of SqueezeNext network is expected with citations
• The abbreviation used first-time should contain long form.
• Figure 8 samples may contain Phoneme
• Samples after preprocessing/normalization can be shown in the figure
• Table 3 and Table 4 contains same captions?
• Structure/Unstructured pruning discussion should be evidenced.

Validity of the findings

• The results should be compared with the state of art work on referred dataset

• The statement in conclusion section- ‘negative samples do not help to improve the performance of the model’ is not empirical.

Additional comments

The work represented for Reconstructed SqueezeNext with C-CBAM for offline
handwritten Chinese character recognition.
The suggested modifications improved the results but not significantly.

---

## Round 0.2 · accepted · Accept

This is an editorial acceptance; publication is dependent on authors meeting all journal policies and guidelines.

Reviewer 1 ·

Basic reporting

The authors present an enhanced, lightweight HCCR model to improve computational efficiency. This is achieved through the reconstruction of SqueezeNext's basic modules for pruning compatibility and the introduction of the Cross-stage Convolutional Block Attention Module (C-CBAM) to improve model performance.

1. The article is written in professional-level English that is easy to understand. However, there are some minor grammatical errors and typos that need to be corrected.

2. As per the suggestion in the last review, the authors have added the citations wherever needed.

3. The Pruning and Quantization sections have been revised, as per the review.

4. Table captions are corrected.

Experimental design

1. The paper proposes to redesign the Basic Module of the SqueezeNext Model to make it compatible with model compression techniques. The experiment is well-explained, and the results demonstrate an improvement in accuracy in this setting.

2. The results demonstrate increased accuracy compared to basic models. To validate the effectiveness of the suggested orders of CAM and SAM, the authors have extensively performed ablation studies.

3. The authors have performed experiments to check the effectiveness of different activation functions as requested in the review.

4. An end-to-end figure has been added as requested.

5. A table has been added to compare their work with prior works, as asked in the review before.

Validity of the findings

The study appears to be replicable and presents potential benefits for Chinese character recognition in low-resource settings. The comparison with prior works is satisfiable.

Reviewer 2 ·

Basic reporting

Satisfactory

Experimental design

Satisfactory

Validity of the findings

Satisfactory

Additional comments

Authors have addressed the comments satisfactorily in the revision.
The revised manuscript can be accepted for publication after the following corrections:
1. The results in Table 7 should include citations for the result comparision
2. In conclusion section, line no 503- bullets used should be removed.